# CONCEPT ATTRACTORS IN LLMS AND THEIR APPLICATIONS

**Sotirios Panagiotis Chytas**     **Vikas Singh**

University of Wisconsin-Madison
chytas@wisc.edu, vsingh@biostat.wisc.edu

## ABSTRACT

Large language models (LLMs) often map semantically related prompts to similar internal representations at specific layers, even when their surface forms differ widely. We show that this behavior can be explained through Iterated Function Systems (IFS), where layers act as contractive mappings toward concept-specific Attractors. We leverage this insight and develop simple, training-free methods that operate directly on these Attractors to solve a wide range of practical tasks, including **language translation**, **hallucination reduction**, **guardrailing**, and **synthetic data generation**. Despite their simplicity, these Attractor-based interventions match or exceed specialized baselines, offering an efficient alternative to heavy fine-tuning, generalizable in scenarios where baselines underperform.

## 1 INTRODUCTION

Consider three distinct concepts: the Lord of the Rings universe, the Python programming language, and 19th-century romantic literature. When prompts from these concepts are given to a large language model (LLM) such as Llama 3.1 Grattafiori et al. [2024], we see an interesting phenomenon. For each concept, despite lexical variations among its prompts, their intermediate representations appear to collapse to distinct regions at *specific layers* – at which layer this happens varies based on the concept. For instance, prompts such as "Who is Gandalf the Grey?" and "What is the significance of Mount Doom?" share minimal similarity on the surface, yet their representations converge to nearly identical locations at layer 24. We see a similar behavior for Python-related queries such

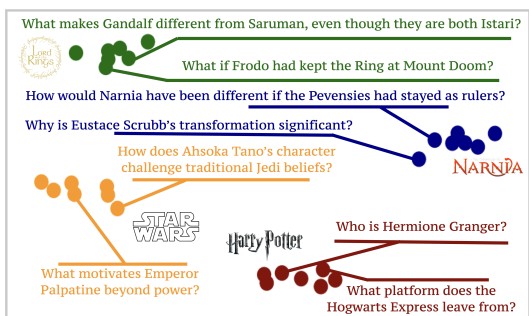

Figure 1: A t-sne van der Maaten & Hinton [2008] plot of the latent representations of Llama3.1-8B for $7 \times 4 = 28$ different prompts, seven each, for the Lord of the Rings universe, Narnia, Star Wars, and Harry Potter. Although the prompts explore different aspects of the universes and share almost no common keywords, we observe a clear clustering based on the different worlds.

as "Help me implement a binary search tree in Python" versus "How can I find the longest non-repeating substring in Python?" and for prompts for the same genre in literature: "Discuss themes in Pride and Prejudice" and "Any easy way to recognize Byron's poetry?". Such a semantic collapse has been reported in some recent results. For instance, Shai et al. [2024] notes that transformer models develop a structured latent representations that encode *belief states*. Separately, Fernando & Guitchounts [2025] suggests that due to the internal dynamics of the model, representations converge to "stable" configurations. From a more practical perspective, Hendel et al. [2023]; Liu et al. [2024d]; Skean et al. [2024] showed that transformers and LLMs shape their latent space according to the underlying task. These findings, while restricted to smaller models and/or for specific contexts, cumulatively support the idea of representation collapse.

A natural question is whether this concept-specific collapse is implied as a property of some underlying dynamical system already studied in the literature, and if so, what guidance can these existing results provide? Specifically, can we obtain strategies for important downstream use-cases? If $p_1, \cdots, p_n$ are a set of prompts related to a specific concept $\mathcal{C}$, we conjecture that the layers of our model may be

acting like a dynamical system that maps semantically related inputs to proximal regions, regardless of their form at the "surface". In other words, the full sequence of layers (leading up to where the representations collapse), if viewed as a unit, implements an iterative (contractive) mapping process to an *Attractor set*, one for each concept. We will see shortly that – to the extent that our hypothesis holds – how existing results are consistent with this view of the collapse phenomena.

**Contributions.** We show that viewing the LLMs through the lens of Iterated Function Systems Barnsley [1988]; Hutchinson [1981] offers a meaningful (or at worst, plausible) explanation for both the layer-specific concept clustering and the subsequent generative process. The main practical benefit is that for a wide-variety of downstream tasks, which are often handled piecemeal in the literature, we can obtain a generic scheme that operates under the assumption that operating with the Attractors alone is *sufficient*. We demonstrate that careful interventions on Attractors can provide us lightweight, *training-free* solutions to a wide array of problems, from **programming language translation** and **guardrailing**, to **hallucination reduction** and **synthetic data generation**. Despite the simplicity as well as limited data/compute needs, these solutions turn out to be comparable to existing specialized approaches. Our experiments focus on Llama3.1 8B Grattafiori et al. [2024]. However, we see a similar behavior on other LLM families too (in particular, Gemma Team et al. [2024] and Qwen Team [2024]), but avoid an exhaustive analysis of all LLMs.

## 2    ITERATED FUNCTION SYSTEMS AND LLMS

There is mounting evidence that large language models (LLMs) possess emergent capabilities beyond simple rote memorization and statistical pattern matching Bender et al. [2021]. Among the many phenomena observed in these models – from in-context learning Dong et al. [2024] to compositional reasoning Lu et al. [2023]; Li et al. [2024b] – we focus on a particular representation-convergence property. Our scope is specifically the collapse phenomena at *specific* intermediate layers. To understand this behavior through the lens of dynamical systems, we hypothesize that LLMs implicitly implement a collection of Iterated Function Systems (IFS) during forward propagation through the layers (Fig. 2).

### 2.1    LLMS IMPLEMENT ITERATED FUNCTION SYSTEMS?

Empirically, we see that for prompts $p_i$, $p_j$ in each concept $\mathcal{C}$, there exists a layer $l$ where:

$$\lim_{l \to l_{\mathcal{C}}} \frac{1}{n^2} \sum_{i,j=1}^{n} |h_l(p_i) - h_l(p_j)| \ll \frac{1}{n^2} \sum_{i,j=1}^{n} |h_0(p_i) - h_0(p_j)| \tag{1}$$

with $h_l$ denoting the implicit transformation by the LLM up to layer $l$. This "squashing" of inter-prompt distances suggests that a contractive mapping process is taking place through the layers. Our hypothesis is that this can be understood via the framework of Iterated Function Systems (IFS) Barnsley [1988]; Hutchinson [1981].

An IFS is defined as a finite set of contractive mappings on a complete metric space. The collective action of these mappings, defined by the Hutchinson operator Hutchinson [1981] is:

$$\mathcal{F}(\mathbf{S}) = \bigcup_{i=1}^{N} f_i(\mathbf{S}) \tag{2}$$

and induces a compact invariant set i.e., $\mathcal{F}(\mathbf{S}^*) = \mathbf{S}^*$, which is called the Attractor of the IFS. More generally, for any initial non-empty compact set $\mathbf{S}_0 \in \mathbb{X}$, the sequence $\{\mathbf{S}_0, \mathbf{S}_1 := \mathcal{F}(\mathbf{S}_0), \mathbf{S}_2 := \mathcal{F}(\mathbf{S}_1), \cdots\}$ converges to $\mathbf{S}^*$ in the Haussdorf metric. More generally, an Attractor in

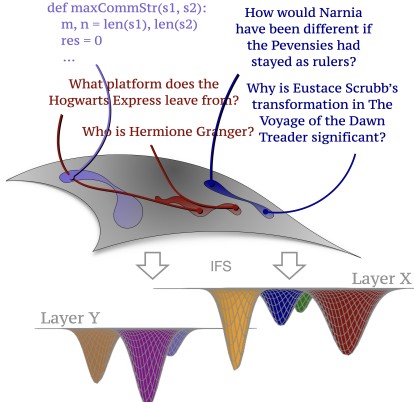

Figure 2: An LLM can be viewed as an IFS that transforms the non-linear manifold of texts into a well-behaving collection of Attractors.

a dynamical system is a closed invariant set toward which trajectories from a wide class of initial conditions evolve asymptotically within its basin of attraction, and may take the form of fixed points,

periodic orbits, tori, or other Attractors characterized by sensitive dependence on initial conditions Barnsley [1988].

Dynamical systems often exhibit Attractors—sets toward which trajectories converge. Simple systems satisfying Banach's fixed-point conditions Banach [1922] converge to a single point, while others yield more complex structures like limit cycles or strange Attractors Strogatz [2024]. We hypothesize that the iterative application of layer transformations in an LLM induces concept-specific invariant sets –semantic Attractors ($\mathbf{A}_l^{\mathcal{C}}$) for each concept $\mathcal{C}$– within the latent space at layer $l$. These compact regions characterize specific concepts, with convergence potentially occurring at different depths depending on the concept.

Once a sequence's representation enters $\mathbf{A}_l^{\mathcal{C}}$, it is further processed by the remaining layers and output matrix $W_{\text{out}}$ to yield a token distribution. Each Attractor may have an invariant measure $\mu_l^{\mathcal{C}}$, describing the distribution of states within it under stochastic dynamics (e.g., varied inputs aligned with concept $\mathcal{C}$). While $\mu_l^{\mathcal{C}}$ is useful for tasks like *synthetic data generation*, it does not directly define next-token probabilities in autoregressive inference, which depend on the specific input-driven state.

The attractors, $\mathbf{A}_l^{\mathcal{C}}$, are linked to the LLM's operational prefill and decode stages. During prefill, the LLM's composed layer transformations guide initial representations of an input prompt, $h_0(p)$, towards $\mathbf{A}_l^{\mathcal{C}}$, with the representation $h_l(p)$ landing within this attractor to give the initial semantic context. Then, during decode, each incremental update to the context (by newly generated tokens) is processed by these same underlying layer dynamics. For coherent generation aligned with concept $\mathcal{C}$, the evolving sequence representation at layer $l$ is continually guided towards or kept within the basin of attraction of $\mathbf{A}_l^{\mathcal{C}}$. Thus, $\mathbf{A}_l^{\mathcal{C}}$ acts like a stabilizing latent structure.

**Collage theorem.** Our operational model takes the transformation performed by the LLM for a concept and approximates it by repeatedly iterating a single affine contractive map Balestriero & Baraniuk [2021], $\phi_{\text{eff}} = M_{\text{eff}}V + t_{\text{eff}}$ (with $V$ as a placeholder hidden representation), suggesting that the overall transformation, for a specific concept, can be roughly approximated by an iterated affine dynamics. We want to estimate the parameters (i.e., the matrix $M_{\text{eff}}$ and vector $t_{\text{eff}}$) and the number of iterations `iter`, that best reproduce the observed mapping (Figure 3).

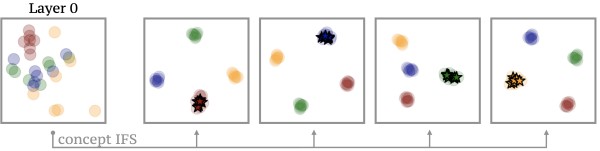

Figure 3: 4 different concepts in layer 0 (before any application of the underlying IFS, and one of the contractions of the underlying IFS we recover by solving the inverse problem for each concept separately. The circles correspond to the true vectors as obtained from the LLM in layer 24 and the stars correspond to the application of the contractions to the points in layer 0.

This is achieved by minimizing the discrepancy between the LLM's observed states at the Attractor layer and the states predicted by iterating $\phi_{\text{eff}}$ from the initial prompt representations:

$$\min_{M_{\text{eff}}, t_{\text{eff}}, \texttt{iter}} \sum_{j=1}^{N} \mathcal{D}\left(h_l(p_j), \phi_{\text{eff}}^{\texttt{iter}}(h_0(p_j))\right) \tag{3}$$

subject to $M_{\text{eff}}$ being contractive (e.g., its operator norm $|M_{\text{eff}}|_{op} < 1$). We apply this `iter` times, and $\mathcal{D}$ is a suitable distance metric. This single map $\phi_{\text{eff}}$ defines a simple Iterated Function System (IFS). The unique Attractor of this 1-map IFS is its fixed point, $V^*$ to which all trajectories $\phi_{\text{eff}}^k(V)$ (for any initial $V$) converge as $k$ grows. The observed empirical set $\mathbf{A}^{\mathcal{C}}$ is then interpreted as the collection of states reached after `iter` applications of $\phi_{\text{eff}}$ starting from the initial set $S_0$. If, as empirical evidence for many concepts suggests, this 1-map model provides a good first-order approximation, then $\mathbf{A}^{\mathcal{C}}$ would be expected to lie in the vicinity of $V^*$. The Collage Theorem Barnsley [1988] states that if $\mathbf{A}^{\mathcal{C}}$ is indeed close to the true Attractor $V^*$ of our fitted $\phi_{\text{eff}}$, then $\mathbf{A}^{\mathcal{C}}$ should be well "collaged" by $\phi_{\text{eff}}$ itself; i.e., $d\left(\mathbf{A}^{\mathcal{C}}, \phi_{\text{eff}}(\mathbf{A}^{\mathcal{C}})\right)$ should be small. While the iterated single affine map is simple, for concepts whose empirical Attractors $\mathbf{A}^{\mathcal{C}}$ exhibit more complex geometries (e.g., disjoint sets or intricate fractal structures not well approximated by convergence to a single point), a richer effective IFS comprising multiple affine maps might be necessary. This would involve finding $\phi$'s and an iteration count `iter`$'$ that minimize $d\left(\mathbf{A}^{\mathcal{C}}, \mathcal{F}^{\texttt{iter}'}(S_0)\right)$, where $\mathcal{F}$ is the Hutchinson operator for the candidate set of $\phi$'s. Alternatively, one could model the geometry of $\mathbf{A}^{\mathcal{C}}$ directly by finding an IFS whose intrinsic Attractor matches $\mathbf{A}^{\mathcal{C}}$, by minimizing the collage error. These approaches are more involved but grounded in IFS theory.

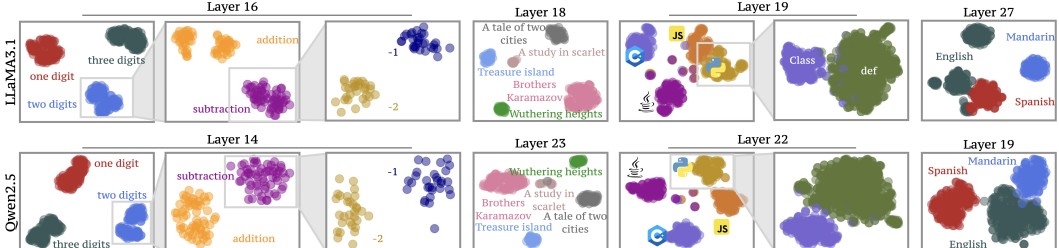

Figure 4: Attractors in Llama3.1-8B Grattafiori et al. [2024] and Qwen2.5-7B Yang et al. [2024]; Team [2024]. From the fractal-like structure of the task vectors in layer 16/14, to literature-based Attractors in layer 18/23 and programming-based in layer 19/22, the treatment of an LLM as an IFS allows us to recover (and use) them in multiple applications, invariant to the underlying LLM. (More models can be found in Section B)

**Does this perspective add to existing results?** Several recent results have indirectly hinted at the IFS-like nature of the LLMs, and more generally transformers, for specific tasks, datasets, and architectures. Fernando & Guitchounts [2025] describes how the intermediate layers of an LLM converge to different "Attractor" points/vectors as the context window of the LLM increases. The result in Wang et al. [2025] examines the Attractors formed in the output layer of an LLM, discovering that paraphrasing results in 2-period cycles. The authors in Shai et al. [2024] present evidence that transformers develop internal representations corresponding to "belief states" over hidden variables in the data-generating process. This phenomenon mirrors the behavior of an IFS, belief states in Shai et al. [2024] can be viewed as specific points within concept Attractors that encode probabilistic information about possible continuations. Notice that the fractal structures reported in Shai et al. [2024] arises naturally from known properties of IFS: systems whose repeated application to an initial set converges to a unique invariant set with so-called *self-similar* properties.

## 2.2 A PRELIMINARY INVESTIGATION OF ATTRACTORS

Before evaluating their practical utility, we first examine the nature of Attractors and their underlying IFS across various concepts and datasets as a sanity check. Unless otherwise stated, we calculate a concept's Attractor value as the average vector representation of all the samples' hidden states for the particular layer. (see Section A)

**Induced tokens.** To understand what the Attractors represent, we average the vectors for each of the four fictional worlds from Fig. 1 to approximate their Attractor points, then project them to vocabulary space via the LLM's final linear layer. The top induced tokens (Table 1) support our hypothesis, revealing meaningful associations—including tokens not present in the original texts, such as the pound symbol (£), filming locations (Auckland, NZ),

Table 1: Top induced tokens of Attractors.

| Concept | Tokens |
|---|---|
| Harry Potter | Harry, wizard, Hogwarts, magical, Voldermort, London, British, £ |
| Lord of the Rings | Lord, Tolkien, Middle, Auckland, NZ |
| Narnia | Kingdom, Tolkien, British, Oxford, Aslan |
| Star Wars | Imperial, Star, galaxy, Galactic, Jedi, Empire, Skywalker, Force, powerful |

or author connections (C.S. Lewis and J.R.R. Tolkien). This suggests the Attractors capture the underlying "essence" of each world, beyond surface-level content.

**Different concepts, different layers.** While for functional worlds, as in Figure 1, we see that the LLM forms clear Attractors in layer 24, this is not the case for all families of concepts, and not discussed in many existing results. We will see later that different families of concepts form Attractors in different layers. For example, we observe the same behavior in layer 19 for programming languages, in layer 27 for natural languages, and in layer 18 for literature books (Figure 4).

**Same concept, multiple Attractors.** Previously, we modeled each concept as a single Attractor (or Concept Vector) in the LLM's latent space. However, some concepts may decompose into multiple sub-concepts. For instance, English forms two distinct Attractors when combining datasets with different semantic styles (`https://www.manythings.org/anki/spa-eng.zip`, `https://huggingface.co/datasets/swaption2009/20k-en-zh-translation-pinyin-hsk`; see Figure 4). This fragmentation is even clearer in layer 16, where tasks produce multiple Attractors based on the number of digits per example.

**A fractal-like structure in the Attractors.** In Figure 4 (left), replicating the setup from Hendel et al. [2023], we observe a structure in the Attractors that empirically resembles that of a fractal. At a high

level, Attractors cluster by the number of digits in the examples. Zooming in, subclusters emerge based on task type (addition vs. subtraction), and further divisions align with specific values being added or subtracted. Similarly, the single cluster of Python programs is further divided into two, based on the solution style (object-oriented vs procedural). This hierarchical structure aligns with theoretical findings in Shai et al. [2024], suggesting a fractal organization of Attractors in this setting. A complete analytical characterization of this phenomenon remains beyond reach with conventional theoretical tools (e.g., box-counting Feldman [2012]). The empirical analysis, however, supports the view that LLMs appear to operate in practice according to this fractal hypothesis.

**LLMs and World Models.** There is much discussion related to whether LLMs operate with an explicit, internal world model Ha & Schmidhuber [2018]. Based on the empirical analysis described so far, we find that there is at least partial evidence to support the idea that the models indeed harbor a *fuzzy* understanding of the world, which is better expressed partially across many of these intermediate layers. In the subsequent section, we will focus on how we can better exploit this fuzzy world model of the LLMs and propose **practical, training free solutions** to a number of use cases.

## 3 ATTRACTOR FOR CONCEPT DETECTION

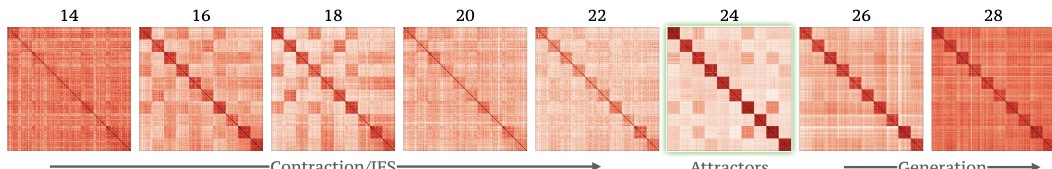

Figure 5: Cosine similarity between all prompts' from TOFU forget05 Maini et al. [2024]. The first 20 rows/columns of each heatmap correspond to questions about the first author, the second 20 about the second author, and so on. The forming of author-based Attractors is apparent and it becomes clearer in layer 24.

Machine unlearning is a active research area, with initial work in computer vision Xu et al. [2023] where many widely used datasets included images of individuals who did not consent to their use. The training datasets of contemporary LLMs are also prompting concern about compliance with the Right to Be Forgotten Chenou & Radu [2019] and similar regulations. Due to the size of these models, retraining or fine-tuning (e.g., Fan et al. [2024]; Jia et al. [2023]; Kurmanji et al. [2023]; Chen et al. [2023]) is often too costly. Moreover, since removal requests are continuous, efficient online unlearning is desirable. To evaluate unlearning in LLMs, Maini et al. [2024] proposed the TOFU benchmark, where models must forget certain fictional authors while retaining performance on others and unrelated tasks.

**Existing solutions.** LLM unlearning methods fall into two main categories: (1) weight reversion and (2) guardrailing. *Weight reversion* seeks new parameters $\theta'$ close to those of a model trained without the forget set, $\theta^*$. Early work Eldan & Russinovich [2023]; Mehta et al. [2022] proposed lightweight fine-tuning to forget specific content (e.g., Harry Potter), but it does not scale to frequent or multi-instance requests. Recent PEFT-based methods Liu et al. [2024h]; Ni et al. [2024] improve efficiency but still require retraining and access to retention data, making them impractical for continuous unlearning. *Guardrailing* avoids changing model weights by intervening at input/output levels. While widely used, such techniques are typically shallow and vulnerable to jailbreaking Jin et al. [2024]; Andriushchenko et al. [2024]. Hybrid approaches like Preference Optimization Maini et al. [2024] use gradient ascent and placeholder outputs but still involve full model fine-tuning and retention data. Other methods (e.g., Liu et al. [2024a]) inject noise using concept classifiers, offering improved efficiency but still need training and retention data for each concept.

**A training-free approach.** We propose a train-free concept guardrailing method for LLMs that requires only data from the concept to be removed – no retention data needed – making it both compute and data efficient. As shown in fig. 5, certain concepts (e.g., TOFU authors) form clear attractors in intermediate layer 24. We estimate each attractor by averaging hidden activations across the concept's samples. At inference, we compute the cosine similarity between the output's attractor and the stored one; if it exceeds a threshold $\tau$, the response is blocked and replaced with a fixed message (e.g., "I cannot provide information about author X due to removal request <id>"). This requires only a single forward pass and no training.

**Evaluation.** Figure 6 (left) shows the cutoff percentage and the model's utility for different values of $\tau$ and for all 3 versions of the TOFU benchmark Maini et al. [2024]. We can observe that even for

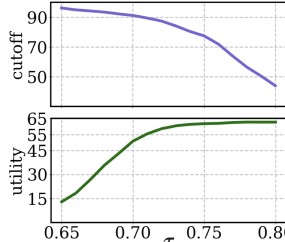

| Method | forget01 | | forget05 | | forget10 | | Train Free | No ret. data |
|---|---|---|---|---|---|---|---|---|
| | Utility ↑ | Rouge ↓ | Utility ↑ | Rouge ↓ | Utility ↑ | Rouge ↓ | | |
| Original | 62.67 | 97.67 | 62.67 | 97.67 | 62.67 | 97.67 | - | - |
| Grad Asc | 60.24 | 43.61 | 00.00 | 00.09 | 00.00 | 00.00 | ✗ | ✓ |
| Grad Diff | 60.59 | 44.80 | 32.44 | 01.85 | 58.23 | 00.32 | ✗ | ✗ |
| Pref Opt | 62.36 | 31.31 | 47.85 | 03.27 | 53.95 | 06.02 | ✗ | ✗ |
| NPO | 45.32 | 24.27 | 17.14 | 19.68 | 17.01 | 20.10 | ✗ | ✓ |
| NPO-RT | 48.96 | 26.55 | 54.14 | 28.93 | 49.97 | 23.80 | ✗ | ✗ |
| ECO | 62.57 | 03.32 | 62.57 | 07.62 | 62.35 | 06.94 | ✗ | ✗ |
| **Ours** | 62.67 | 00.48 | 61.20 | 10.33 | 61.34 | 19.54 | ✓ | ✓ |

Figure 6: (left) Model utility and cutoff as functions of $\tau$ for TOFU forget10 Maini et al. [2024]. Model utility measures the effect of guardrailing on the LLM's general answering ability, while cutoff is the percentage of forget-set questions detected and guardrailed. (right) Model utility and Forget Rouge of our train-free method compared with typical (e.g., Gradient Ascent) and recent trainable methods (e.g., NPO Zhang et al. [2024], ECO Liu et al. [2024b]). Despite requiring no retention data, our approach outperforms most baselines and offers finer control over the tradeoff between model utility and cutoff/Rouge through the parameter $\tau$.

the hardest version (forget10), the model's utility remains high while we enjoy a cutoff percentage of more than $90\%$. For specifically chosen values of $\tau$, we show in Figure 6 (right) that our train-free approach is competitive with many heavier, trainable solutions. At the same time, the use of $\tau$ allows a finer control over the tradeoff of forgetting versus model utility.

**Can two Authors occupy the same latent space?** While there is no theoretical guarantee that two authors cannot share the same latent region—potentially causing guardrailing for one to unintentionally "remove" the other—our experimental findings indicate that this does not occur in practice (see Section A). Additionally, Utility implicitly captures this effect: any unintended removal of facts, people, or places would immediately reduce its value (Section C). In contrast, as shown in Figure 6, our model achieves some of the highest Utility scores among both trainable and train-free methods.

## 4 ATTRACTORS FOR TRAVERSALS

Treating the LLM as an IFS, and more generally a dynamical system, allows us to intervene on its trajectory and guide it towards specific Attractors. From a dynamical system perspective, if we assume that the LLM can be characterized from a function $f$ such that $dx/dt = f(x)$, then, given a target Attractor $y$, we can modify the system as $dx/dt = f(x) + \lambda(y - x)$ and steer it towards another Attractor $y$, with $\lambda$ being influenced by the underlying dynamics of the system (robustness to perturbations, distance of Attractors, etc.).

Such an approach, called *steering*, has been variously studied. We know that carefully chosen vectors can steer a model's behavior so that its output is less toxic, more poetic, etc. Li et al. [2024a]; Liu et al. [2024f]; Beaglehole et al. [2025], essentially steering the model internally to different Attractors. However, many of these approaches require training the model itself or auxiliary smaller networks (e.g., Beaglehole et al. [2025]; Postmus & Abreu [2024]; Huang [2024]), while other works require carefully chosen data that satisfy some, more or less restrictive, assumptions (e.g., Liu et al. [2024f]; Gu et al. [2024]; Song et al. [2025]).

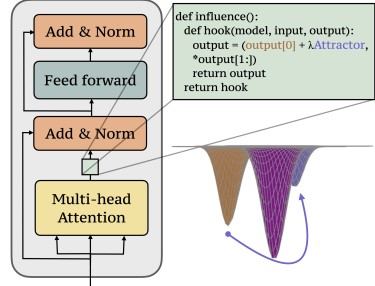

Figure 7: Influencing the dynamics of the LLM by adding the target Attractor. The only modification needed is the introduction of a forward hook on the appropriate layer.

Unlike methods requiring extensive retraining or retention data, we show that simply adding or subtracting Attractors at selected intermediate layers can influence LLM behavior across tasks – from detoxification to code translation – without these constraints. Surprisingly, in practice, the *before* Attractor is mostly unnecessary, removing the need for retention data entirely. Despite requiring only a single forward pass over target data and no training, our approach matches the performance of more resource-intensive methods.

### 4.1 DRIFTING AWAY FROM THE TOXICITY ATTRACTOR

Multiple works have shown that careful manipulation of the activations across the LLM's layers allows us to control its behavior, and a common application is toxicity reduction. We note that these

| Method | Toxicity (↓) | Rouge (↑) | Train free | No ret. data |
|---|---|---|---|---|
| Original dataset | 84.58 | - | - | - |
| Llama2-7B | 71.60 | 73.15 | - | - |
| ICL | 66.81 | 74.19 | ✓ | ✗ |
| LoRA | 49.94 | 57.32 | ✗ | ✗ |
| ICV | 39.54 | 65.97 | ✓ | ✗ |
| **Ours** | **37.66** | 61.27 | ✓ | ✓ |

ok - i can second this , that nicaraguan coffee is f*cking amazing.

I can confirm that the coffee from Nicaragua is excellent.

this sh*t is why i moved to cali and refuse to look back.

This is why I moved to California and will not look back.

lol sh*t i need change.

I need to change my life

for example i don 't give a sh*t about your opinions , yet you still comment.

I don't care about your opinions, but you still commented.

Figure 8: (left) Toxicity score and Rouge on ParaDetox. Although our lightweight approach requires no training or even retention data, it is reducing significantly the toxicity while maintaining the textual quality. (right) Toxic examples and the modified passages according to our method.

ideas impose one or more restrictive requirements on the data format, such as the need for retention data, or even the existence of paired data Liu et al. [2024f]; Li et al. [2024a]. Here, we check whether the estimation of the toxicity Attractor alone allows us steer the generation away from it and thereby, reducing the toxicity content of the LLM's output. No additional assumptions on the data are needed. Using the ParaDetox Logacheva et al. [2022] dataset, we obtain a single vector estimate of the toxicity Attractor on layer 16 and, then, during generation, we subtract this value from each token's activation on layer 16, essentially discouraging the generation to converge to the toxicity Attractor. Although we only require the toxicity Attractor/vector, our targeted approach performs better than many of the existing (but more restrictive) solutions.

**Evaluation.** In Figure 8, we show that our approach, without any need for training/retention data, performs similar as ICV Liu et al. [2024f] which needs a PCA projection of the differences between paired samples. We also appear to perform better than LoRA fine-tuning or the more lightweight In-Context Learning Dong et al. [2024]. To assess both the reduction in toxicity as well as any potential drop in the quality of the generated text, we report both Toxicity Miller et al. [2017], as well as the Rouge score Lin [2004]. Our approach is one of the few training free methods and the only one that requires no retention data. We find that relaxing these requirements does not lead to a performance drop, instead a performance gain. Finally, we should note that there are practical benefits of our lightweight approach.

## 4.2 SWITCHING LANGUAGE ATTRACTOR ON THE FLY

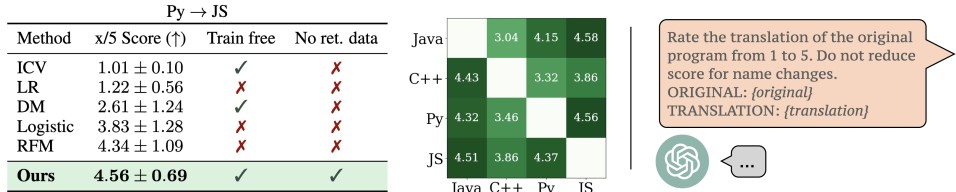

| | Py → JS | | |
|---|---|---|---|
| Method | x/5 Score (↑) | Train free | No ret. data |
| ICV | $1.01 \pm 0.10$ | ✓ | ✗ |
| LR | $1.22 \pm 0.56$ | ✗ | ✗ |
| DM | $2.61 \pm 1.24$ | ✓ | ✗ |
| Logistic | $3.83 \pm 1.28$ | ✗ | ✗ |
| RFM | $4.34 \pm 1.09$ | ✗ | ✗ |
| **Ours** | $4.56 \pm 0.69$ | ✓ | ✓ |

Rate the translation of the original program from 1 to 5. Do not reduce score for name changes.
ORIGINAL: {original}
TRANSLATION: {translation}

Figure 9: (left) LLM as a transpiler. For all pairs of the four considered languages, switching the Attractor to the target language can successfully make the LLM act as a transpiler without any specific such instructions or retention data. (right) Using o4 to judge the quality of the generated translations.

LLMs are extremely capable at code comprehension and composition Fang et al. [2024]; Denny et al. [2024]; Wadhwa et al. [2024]. Other than use as a code-generation assistant, an important use case is as a transpiler, especially for programming languages with limited support. Typically, the approach involves a data-intense stage of fine-tuning on code-specific data (e.g., Roziere et al. [2023]; Li et al. [2022]). Some recent works have evaluated the limits of zero/few-shot transpiling in LLMs Bhatia et al. [2024]; Beaglehole et al. [2025].

As shown in Figure 4, some programming languages form Attractors on layer 19 of Llama3.1-8B. We test whether these Attractors let the LLM act as a transpiler: given only a code block in one language, can it translate to a target language without special instructions? Using 100 LeetCode solutions in Python, Java, C++, and JavaScript, we estimate the layer-19 Attractors. Assuming input code in language X converges to Attractor X, we then examine generation when traversing the Attractor space to the Attractor of another language Y.

**Evaluation.** To evaluate the quality of the generated code, we use o4-judge to provide us with a score of the quality of the generated code in the target language. As shown in Figure 9, we can successfully repurpose the LLM as a transpiler without any demonstrations (zero-shot) as well as no other relevant information in the prompt. We achieve impressive results for all pairs of the 4

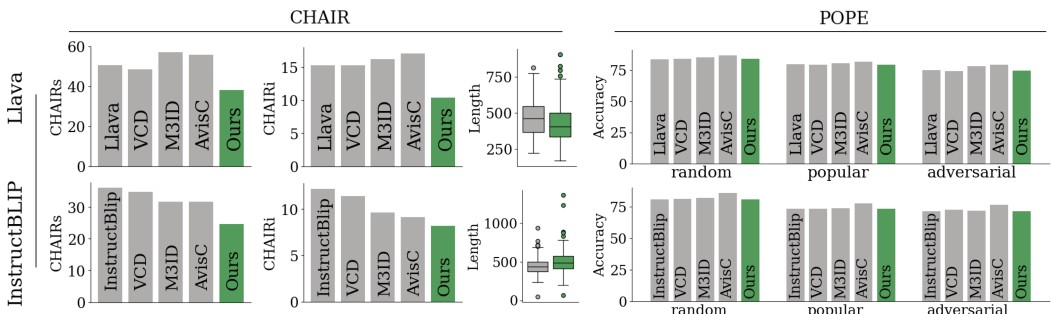

Figure 10: CHAIR Rohrbach et al. [2018] and POPE Yifan Li & Wen [2023] on Llava-1.5 Liu et al. [2024c] and InstructBLIP Dai et al. [2023]. While our approach maintains performance on the discriminative questions (POPE) it significantly reduces the hallucinations in the generative tasks (CHAIR), without affecting the length of the generated descriptions.

considered languages. We do not require any retention data, additional training, or an increase in the inference time. We obtain a score better than other simple, train-free approaches (e.g., Difference of Means (DM) Beaglehole et al. [2025] and ICV Liu et al. [2024f]) as well as approaches that involve training auxiliary classifiers (e.g., RFM, LR Beaglehole et al. [2025]).

### 4.3 REMAINING ON THE VISUAL ATTRACTOR

Hallucinations are a well-known issue in LLMs Martino et al. [2023]; Friel & Sanyal [2023]; Huang et al. [2025], amplified in Vision-Language Models (VLMs) by a fading memory effect where attention to visual input diminishes Favero et al. [2024]; Liu et al. [2024g]. We hypothesize this stems from a shift between Attractors: VLMs start aligned with a visual Attractor but drift toward a text-only Attractor due to LLM pretraining. To counter this, we add the initial visual Attractor vector (computed at the first generation step) to the hidden state at each subsequent step, reinforcing visual grounding. Unlike prior methods (Figure 7), our approach dynamically computes and maintains the visual Attractor throughout generation.

**Evaluation.** Compared to other train-free approaches (e.g., Favero et al. [2024]; Woo et al. [2024]; Leng et al. [2024]), our algorithm does not lead to an increase in inference time, since it does not require multiple forward passes. Despite its simplicity, the results are strong, leading to a significant reduction in the hallucination rate of two widely used VLMs (InstructBLIP Dai et al. [2023] and Llava-1.5 Liu et al. [2024c]), as shown in Figure 10 (CHAIR). Our modification also does not affect the general abilities of the VLM, resulting in a similar (or slightly improved) performance on discriminative questions.

### 5 ATTRACTORS PERTURBATION FOR DATA GENERATION

Recent studies show that LLMs can generate new samples resembling small real datasets. Various works explore prompting strategies and multi-step methods to improve sample quality Long et al. [2024]. Others note the challenge of prompt design and propose minimal fine-tuning to turn an LLM into an autoencoder that produces new samples via high-temperature sampling DeSalvo et al. [2024].

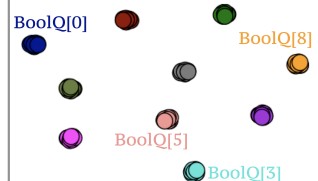

Figure 11: Sample-based Attractors for different generation instructions. Each Attractor corresponds to one sample from BoolQ.

**Limitations of Temperature sampling.** LLM output variability is typically controlled by Temperature and related parameters (top-K, top-P), which add stochasticity. Yet even with high randomness, outputs often remain limited and lack diversity when generating text similar to existing data Gandhi et al. [2024]; Liu et al. [2024e]; Long et al. [2024]. This is usually mitigated through carefully tuned or multiple prompts, but that approach does not scale or suit large-scale synthetic data generation.

A common approach to boost diversity beyond temperature sampling is running multiple forward passes with varied prompts while keeping the same original sample. Studies show that carefully tuned instructions can yield more diverse synthetic outputs Gandhi et al. [2024]; Liu et al. [2024e]; Long

| | BoolQ | | AG | | |
|---|---|---|---|---|---|
| | Qwen2.5-0.5B | GPTNeo-1.3B | Qwen2.5-0.5B | GPTNeo-1.3B | |
| No train | 38.47 | 38.53 | - | - | |
| No augmentation | 62.54 | 62.17 | 30.66 | 23.46 | |
| Temp sampling | 64.16(±2.98) | 64.80(±2.80) | 82.91(±2.58) | 50.96(±19.45) | |
| **Ours** | **69.28(±0.88)** | **67.77(±1.86)** | **85.64(±0.59)** | **72.06(±7.51)** | |

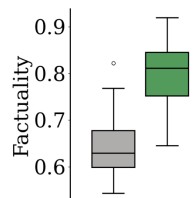

Figure 12: (left) Test-set accuracy on BoolQ and AG when trained with synthetic datasets generated through temperature sampling and our approach. In all cases, our dataset results in a more generalizable model with better performance. (right) Factuality of generated facts about popular figures with temperature sampling (gray) and our approach (green). We observe a more than 20% increase in the factuality on average.

et al. [2024]. However, this demands laborious, non-automated prompt design with trial-and-error and becomes impractical for large, heterogeneous datasets like BoolQ Clark et al. [2019].

## 5.1 ATTRACTOR PERTURBATIONS: REPLICATING THE EFFECT OF MULTIPLE TAILORED INSTRUCTIONS

Similar to previous experiments, we investigate whether sample-wise Attractors exist for diverse instructions. Is there a layer where different instruction trajectories "collapse" for the same sample? Yes—Figure 11 shows that with 10 BoolQ-specific instructions, all trajectories converge on layer 16, forming sample-wise Attractors. Building on this, we test whether perturbing the Attractor estimated from a single instruction can replicate the diversity achieved with multiple prompts. Using only one (possibly simple) prompt, can we generate multiple diverse samples without raising temperature and risking corrupted, nonsensical outputs? As shown later, this simple, train- and tuning-free approach yields higher-quality data, validated through both direct and indirect evaluations.

**Estimating the quality of the generated data.** We evaluate two textual datasets, BoolQ Clark et al. [2019] and AG Zhang et al. [2015]. Although both are relatively large and diverse, we use minimal versions of 100 samples each to reduce the original train set's influence and better assess each generation method. Using these 100 samples, we prompt Llama3.1-8B to generate new synthetic samples via both typical temperature sampling and our approach. To assess the quality of the generated data, we perform indirect evaluation by fine-tuning smaller LLMs on the synthetic collections Long et al. [2024]. Specifically, we use Qwen2.5-0.5B Team [2024] and GPTNeo-1.3B Black et al. [2021]. In Figure 12 (left), we report test accuracy when training each model on each dataset version. The quality improvements are clear, yielding better results in all cases.

**Estimating the factuality of the generated data.** Besides the indirect comparison, we also evaluate the generated samples' quality directly. Following Tian et al. [2024], we prompt the model to produce facts for a collection of randomly selected celebrities and historical figures. To assess factuality we use o4-judge, prompting it to label each generated fact as `true` or `false`. In Figure 12 (right) we show that factuality is much lower with temperature sampling; using Attractors yields an absolute increase of 20% on average. Detailed per-person improvements are reported in the appendix.

## 6 CONCLUSION

This work is based on the hypothesis that the evolution of hidden representations of prompts in Large Language Models (LLMs), specifically their convergence to distinct internal representations (for semantically related prompts), can be understood through the framework of Iterated Function Systems (IFS). We check that LLM layers progressively map inputs towards concept-specific "Attractors" in their latent space. Building on this perspective, we evaluated a range of simple, training-free ideas that directly manipulate these identified Attractors. On a diverse set of practical tasks, including machine unlearning (guardrailing against specific concepts), guiding LLM generation for tasks like code translation and toxicity reduction, mitigating hallucinations in vision-language models, and improving the diversity and factuality of synthetic data generation, we find that our proposal offers surprisingly strong performance. It is computationally efficient and there is no need of re-training or fine-tuning, and offers a clear and promising direction for evaluating applicability in other use-cases.

**Impact & Limitations.** Modeling LLMs as IFS can yield solutions to diverse problems and potentially extend their capabilities. A key limitation is the need for direct access to hidden activations to estimate and manipulate the concept Attractors, which standard black-box APIs do not provide. Due to computational limits, we evaluated models up to 8B parameters, leaving it to future work to test whether similar Attractor phenomena appear in larger models.

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

## A   ATTRACTORS ESTIMATION

In all of our experiments, we estimate the concept's Attractor as the mean value of the hidden state of all samples (e.g., all toxicity samples or all author questions) for the Attractors' layer. The layer is chosen as the one that maximizes the inter-distances while minimizing the intra ones. More specifically, assume $M * N$ prompts, where each set of $M$ prompts corresponds to a specific concept (e.g., Harry Potter). For each layer $l$, we calculate the concept's Attractor as

$$\mathbf{a}_i^{(l)} = \frac{1}{M} \sum_{j=0}^{M-1} \mathbf{h}_{i*M+j}^{(l)}, \quad \forall i \in [1, N] \tag{4}$$

where $\mathbf{h}_k^{(l)}$ corresponds to the hidden state of prompt $k$ at layer $l$. Based on the estimated Attractors, we calculate two different metrics: (1) inter-distance, and (2) intra-distance. The inter-distance is defined as the average distance between all the different pairs of Attractors, i.e.,

$$\text{inter}^{(l)} = \frac{1}{\frac{M^2-M}{2}} \sum_{i=1}^{M} \sum_{j=i+1}^{M} \mathcal{D}(\mathbf{a}_i^{(l)}, \mathbf{a}_j^{(l)}) \tag{5}$$

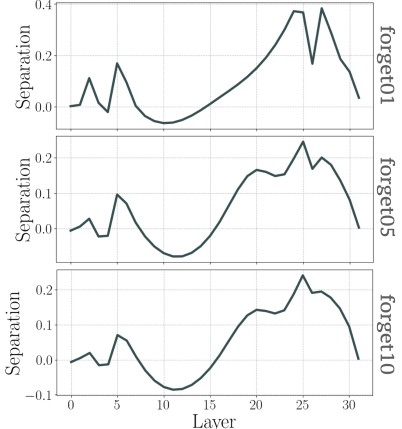

Figure 13: Separation per layer for all three forget sets of TOFU. Interestingly, the optimal layer does not change between different forget sets, strengthening the IFS assumption that a single layer is "responsible" for all authors' Attractors.

while the intra-distance is calculated as the average distance of each hidden state for its corresponding Attractor (i.e., the spread):

$$\text{intra}^{(l)} = \frac{1}{M * N} \sum_{i=1}^{N} \sum_{j=0}^{M-1} \mathcal{D}(\mathbf{a}_i^{(l)}, \mathbf{h}_{i*M+j}^{(l)}) \tag{6}$$

where $\mathcal{D}$ is an appropriately chosen distance measure (cosine similarity in our experiments). Based on these measurements, we define Separation as

$$\text{Separation}^{(l)} = \text{inter}^{(l)} - \text{intra}^{(l)} \tag{7}$$

and chose as the ideal layer $l^*$ the layer that maximizes this quantity:

$$l^* = \arg\max_l \text{Separation}^{(l)} \tag{8}$$

In practice, we observe, as expected by the IFS theory, the layer choice is quite clear even by simply examining the accompanying t-sne/distance plots (e.g., Figures 4 and 5). In Figure 13 we depict the Separation we obtain for each forget set, results that agree with Figure 5.

## B   ATTRACTORS ACROSS MODEL SIZES

In Figure 4 we demonstrated that contemporary LLMs, like LlaMA3.1 Grattafiori et al. [2024] and Qwen2.5 Team [2024], form attractors in different layers for many and diverse concepts, from mathematical operations, to literature and programming languages. However, our experiments were based on a specific size (7-8B) casting doubts on the generality of the results. In Figure 14 we demonstrate the IFS nature for a widecollections of LLM sizes, spanning from 3B to 14B models. Our initial observations hold in this expanded family of models too, showcasing that our approach can be used in practice for more or less powerful LLMs, depending on someone's needs and capabilities.

## C   ATTRACTOR FOR CONCEPT DETECTION

**Data.**   The TOFU benchmark Maini et al. [2024] uses a synthetic dataset crafted to test how well LLMs can forget specific information. It features 200 made-up author profiles, each with 20 question-answer pairs detailing aspects like birthplace, genre, and awards. These profiles were generated using

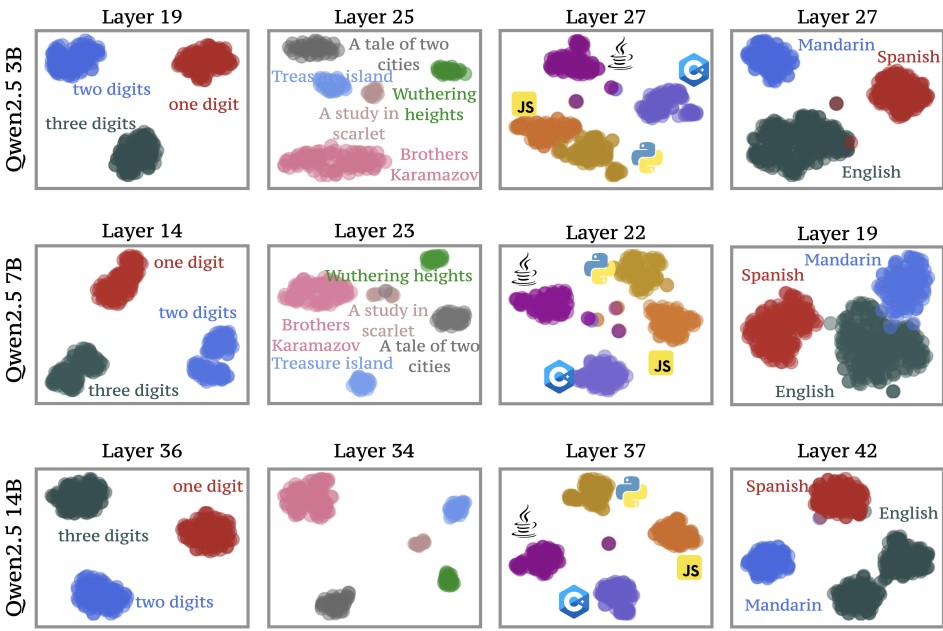

Figure 14: Attractors in Qwen2.5 Qwen2.5-7B Yang et al. [2024]; Team [2024] family of models, from 3B to 14B. Similarly to Figure 4, the formation of attractors is aparent on all models, although on different layers for each model.

GPT-4, ensuring they don't exist in any real-world data. To evaluate unlearning, a subset of these profiles, called the "forget set", is designated for the model to forget. Three different variations were introduced –forget01, forget05, and forget10– that correspond to different percentages of the authors to be forgotten. The rest form the "retain set" which the model should remember. Additionally, TOFU includes evaluation datasets with real authors and general world facts to assess whether unlearning specific information affects the model's broader knowledge.

**Models.**    Using the fictitious data from above, TOFU then finetuned multiple LLMs on different subsets of them. One was trained on everything but forget10, one in everything but forget05, one in everything but forget01, and finally one was trained on the whole dataset. The fully-trained model is the one used to test different unlearning methods, while the three partially-trained models correspond to the ideal models and parameters ($\theta^*$) that unlearning methods seek.

**Evaluation metrics.**

1. **Probability**: The Probability metric assesses the model's confidence in generating the correct answer $a$ given a question $q$. To normalize for answer length, the probability is adjusted as follows:
$$P(a \mid q)^{1/|a|} \tag{9}$$
where $|a|$ denotes the number of tokens in the answer. This normalization ensures fair comparison across answers of varying lengths.

2. **ROUGE-L Recall Score**: The ROUGE-L Recall Score measures the overlap between the model's generated answer and the ground truth answer, focusing on the longest common subsequence. It captures the model's ability to produce answers that are similar in content and structure to the expected responses, even if the wording differs.

3. **Truth Ratio** The Truth Ratio compares the model's confidence in a paraphrased correct answer $\tilde{a}$ to its confidence in several perturbed (incorrect) versions $\hat{a} \in A_{\text{pert}}$. It is defined as:
$$R_{\text{truth}} = \min\left( \frac{1}{|A_{\text{pert}}|} \sum_{\hat{a} \in A_{\text{pert}}} \frac{P(\hat{a} \mid q)^{1/|\hat{a}|}}{P(\tilde{a} \mid q)^{1/|\tilde{a}|}}, \frac{P(\tilde{a} \mid q)^{1/|\tilde{a}|}}{\frac{1}{|A_{\text{pert}}|} \sum_{\hat{a} \in A_{\text{pert}}} P(\hat{a} \mid q)^{1/|\hat{a}|}} \right) \tag{10}$$

This metric reflects the model's ability to distinguish correct answers from incorrect ones. A lower Truth Ratio indicates better unlearning performance.

4. **Model Utility**: The utility score of the model is derived from the harmonic mean of nine individual measures: answer probability, truth ratio, and ROUGE recall for each of the three evaluation subsets –retain, real authors, and world facts. A higher utility score is indicative of better model performance.

5. **Forget Cutoff**: This metric is introduced by us, and it is depicted in Figure 6 (left). Since our method is about guardrailing specific authors, we are interested in the percentage of author-related questions that are correctly detected (and cutted off).

**Baselines.** The complete details on all baselines can be found in Liu et al. [2024b].

## D ATTRACTORS FOR TRAVERSALS

### D.1 DRIFTING AWAY FROM THE TOXICITY ATTRACTOR

**Data.** The ParaDetox dataset is a key resource for training models to rephrase toxic language into neutral expressions Logacheva et al. [2022]. It comprises over 10,000 English sentence pairs, each featuring a toxic sentence and its non-toxic paraphrase. The dataset was created through a structured crowdsourcing process on Toloka.ai, involving paraphrasing, content preservation checks, and toxicity verification. This approach ensured high-quality data for developing effective detoxification models.

**Baselines.** In Figure 8 we compared our method against 3 different baselines. Here is a breakdown of each one:

1. **ICL**: ICL, which stands for In-Context Learning Dong et al. [2024], utilizes the LLM with specific prompts and a few examples (demonstrations) to guide detoxification without altering model weights.

2. **LoRA**: LoRA, which stands for Low Rank Adaptation Hu et al. [2022], finetunes the model on the specific dataset (ParaDetox Logacheva et al. [2022]). Although the "heaviest" of all methods, since it evolves training (some) of the LLM's parameters, the results are not better than more lightweight approaches, like ours.

3. **ICV**: ICV, which stands for In-Context Vectors Liu et al. [2024f], calculates a "task vector" using a small set of (paired) in-context examples. This vector encapsulates the task's essence and is used to modulate the model's behavior for detoxification tasks without additional fine-tuning.

### D.2 SWITCHING LANGUAGE ATTRACTORS ON THE FLY

**Data.** To estimate the programming languages Attractors we used solutions from LeetCode's problems from `https://huggingface.co/datasets/greengerong/leetcode`. Each sample of the dataset consists of the question and its difficulty, as well as the corresponding solutions in Python, Java, C++, and Javascript.

**Baselines.** In Figure 9 we compared our method against 5 different baselines. Here is the details of each one:

1. **ICV**: ICV, which stands for In-Context Vectors Liu et al. [2024f], calculates a "task vector" using a small set of (paired) in-context examples. In Beaglehole et al. [2025] it can be also found as "PCA".

2. **Logistic Regression**: A linear classifier applied to the activations of a single layer within the LLM. It serves as a baseline in Beaglehole et al. [2025] to assess the effectiveness of simple linear decision boundaries in detecting specific concepts.

3. **Linear Regression**: Similar to logistic regression, the underlying classifier in this case is linear regression.

4. **Diference of Means (DM)**: A method that involves directly matching the hidden representations corresponding to specific concepts without any learned transformation.

5. **Recursive Feature Machine (RFM)**: Beaglehole et al. [2025] novel approach that leverages nonlinear feature learning across multiple layers of an LLM to identify and manipulate semantic concepts. RFM combines features from different layers to build powerful concept detectors and steering mechanisms, demonstrating state-of-the-art results on various benchmarks.

### D.3 Remaining on the visual Attractor

**Benchmarks.** In Figure 10 we demonstrated our approaches superiority in two different benchmarks. Each one evaluates a different hallucination aspect and the details can be found below:

1. **POPE**: POPEYifan Li & Wen [2023] –short for Polling-based Object Probing Evaluation– is a tool designed to assess object hallucination in VLMs. POPE evaluates this by prompting models with simple yes-or-no questions about specific objects in an image (e.g., "Is there a cat in the image?") and comparing the responses to ground-truth annotations. This method provides a straightforward way to quantify hallucination rates across different models and datasets, with the focus being on discriminative questions.

2. **CHAIR**: CHAIRRohrbach et al. [2018], which stands for Caption Hallucination Assessment with Image Relevance, is a metric designed to evaluate object hallucinations in image captioning models. It measures the proportion of objects mentioned in a generated caption that are not present in the corresponding image. This helps in assessing how often a model "hallucinates" objects, i.e., describes items that are not actually in the image.

   The CHAIR metric operates at two levels:

   - *Instance-level (CHAIRi)*: Calculates the percentage of hallucinated object instances relative to all object instances mentioned in the caption.
   - *Sentence-level (CHAIRs)*: Determines the percentage of sentences that contain at least one hallucinated object.

   By analyzing both levels, CHAIR provides a comprehensive view of a model's tendency to hallucinate objects in image captions. It has been widely adopted in the evaluation of vision-language models, especially when assessing their performance on datasets like MSCOCO Lin et al. [2014].

**Baselines.** We consider three contemporary, train-free methods for hallucation reduction. In contrast to our approach, these methods require multiple inference passes, increasing the generation time for each new query.

- **VCD**: VCD Leng et al. [2024] operates as a training-free technique that modifies the decoding process during inference. It contrasts the model's output distributions when provided with the original image versus a deliberately distorted version of the same image. The core idea is that by comparing these outputs, the model can identify and suppress content that is overly influenced by language priors rather than the actual visual input.

- **M3ID**: M3ID Favero et al. [2024] addresses the issue of hallucinations by maximizing the mutual information between the generated text and the visual input. The method operates during inference and can be applied to any pre-trained autoregressive LVLM without additional training. By focusing on enhancing the alignment between visual and textual modalities, M3ID encourages the model to generate outputs that are more grounded in the visual content.

- **AvisC**: AVISC Woo et al. [2024] addresses hallucinations by analyzing and adjusting the attention distribution over visual tokens during the decoding phase. The method identifies "blind tokens", which are tokens that receive disproportionately low attention weights yet may contain critical visual information. By contrasting the model's output logits conditioned on the original visual tokens with those conditioned on the blind tokens, AVISC dynamically adjusts the logits to reduce the model's dependency on blind tokens. This encourages a more balanced consideration of all visual tokens, leading to outputs that are more grounded in the visual content.

**Additional results.** Figure 15 depicts the impact of re-enforcing the visual Attractor on Llava1.5 Liu et al. [2024c]. In all cases, we are able to eliminate the hallucinations of the unmodified model, without introducing new ones.

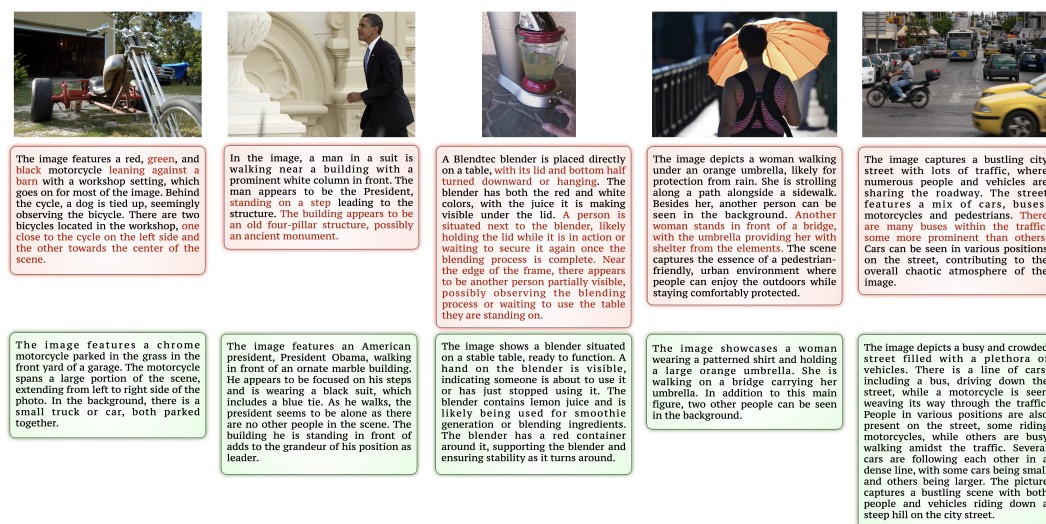

Figure 15: Before and after re-enforcing the visual Attractor, on Llava1.5 Liu et al. [2024c]

# E    ATTRACTORS PERTURBATION FOR DATA GENERATION

**Datasets.** Our experiments deal with the following two datasets. Despite their quite large size, to better assess the quality of the synthetically generated data, we considered only a small subset of 100 real samples.

1. **BoolQ**: BoolQ Clark et al. [2019] is a benchmark designed for evaluating reading comprehension systems on yes/no questions. The dataset comprises 15,942 examples, each consisting of a naturally occurring question, a passage from a Wikipedia article, and a boolean answer (true or false). These questions are not artificially generated; instead, they are real queries posed by users, making the dataset particularly valuable for assessing models in realistic scenarios.

   Each sample in BoolQ includes:

   (a) *Passage*: A segment of text from a Wikipedia article that contains information relevant to the question.

   (b) *Question*: A naturally occurring yes/no question that can be answered solely on the information provided on the passage.

   (c) *Answer*: A boolean value indicating the correct answer to the question based on the passage.

   The questions in BoolQ often require complex reasoning and understanding of the passage, making it a challenging benchmark for models.

2. **AG**: The AG News dataset is a subset of the AG's corpus of news articles Zhang et al. [2015]. It was constructed by selecting articles from the four largest categories in the original corpus: (a) *World* (b) *Sports* (c) *Business* (d)  . Each article in the dataset includes a title and a short description, providing concise textual content for classification tasks.

**Prompting.** To generate the synthetic samples, we prompted Llama3.1-8B Grattafiori et al. [2024] 10 times for each sample. The prompts used for each dataset can be seen below:

> **BoolQ**: <sample>. Now generate 3 different passages, questions, and answers similar to the example above. Please make sure each question you generate has a boolean answer that can be answered by the passage. Make sure each passage and question is different and sufficiently rephrased. Please make sure you generate passages, questions and both true and false answers.

> **AG**: <sample>. Now generate 3 different texts and their corresponding class similar to the example above. Make sure each text is not too long and it is different and sufficiently rephrased. Please make sure each class you generate belongs to one of the four classes (Technology, World, Business, Sports).

The same prompts were used in both temperature sampling and our, attractor-based, approach.

**Models and hyperparameters.** After obtaining the synthetic data using Llama3.1-8B Grattafiori et al. [2024], we finetune two smaller LLMs (Qwen2.5-0.5B Team [2024] and GPTNeo-1.3B Black et al. [2021]) on them. Table 2 displays all the hyperparameters used in all different trains.

Table 2: Training hyperparameters for both datasets and LLMs.

| Hyperparameter | BoolQ Clark et al. [2019] | | AG Zhang et al. [2015] | |
|---|---|---|---|---|
| | Qwen2.5-0.5B | GPTNeo-1.3B | Qwen2.5-0.5B | GPTNeo-1.3B |
| learning rate | $5e-5$ | $5e-5$ | $5e-5$ | $5e-5$ |
| batch size | 8 | 8 | 16 | 32 |
| max epochs | 10 | 10 | 5 | 5 |

**Factuality estimation.** To assess the factuality of the generated facts of both methods examined, we considered a dataset of 35 distinct famous personalities, such as Nelson Mandela and Pablo Picasso. Using this list, we prompted Llama3.1-8B Grattafiori et al. [2024] 10 times to generate 10 different facts for each person. Using these facts, we employed o4-judge to determine the factuality of each one. On average our method achieves a $20\%$ increase in the factuality, and the indivual increases can be found in fig. 16.

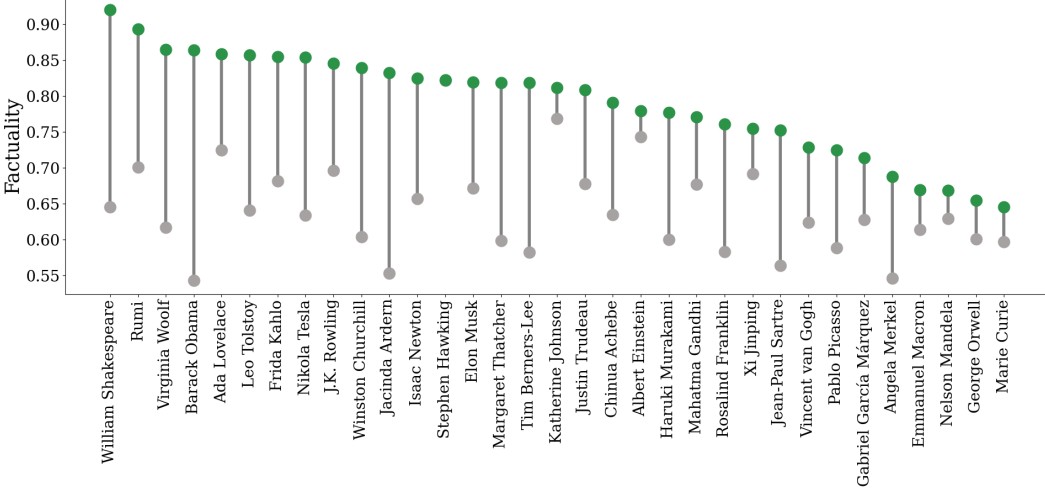

Figure 16: Factuality percentage of temperature sampling (gray dots) and our approach (green dots). The improvement is apparent in all cases, reaching as much as $30\%$.

## F  SAME CONCEPT, MULTIPLE ATTRACTORS

In Figure 4 we demonstrated the fractal nature of the Attractors for multiple concepts (e.g., arithmetic tasks). Here we demonstrate the impact of this hierarchical nature of the Attractors for the arithmetic

task. Following Hendel et al. [2023], we examine whether a model can perform simple arithmetic tasks (e.g, "−2") without any explicit instructions, similar to our experiments of Section 4. More specifically, assuming an appropriately chosen Attractor $\mathbf{a}$ in layer $l$ that corresponds to the concept of an operation (e.g., "−2"), we prompt the model with the following prompt: "i->" (where i corresponds to a number) and, internally, we add $\mathbf{a}$ in its corresponding layer, as we demonstrate in Figure 7. The quality of the intervention is measured as the fraction of numbers in a specific range for which the model correctly predicts the underling operation.

Knowing that an LLM internally forms multiple Attractors for each operation (Figure 4), depending on the number of digits on the demonstrations, we seek to examine not the raw performance of our interventions but rather the relative performance of utilizing different Attractors. Are all the Attractors equally good demonstrations of the task, or the multi-Attractor formation implies that, the more general the Attractor the more noisy (and less effective) it is? In Table 3 we display the results, which clearly show that a simple averaging of all the Length-specific Attractors can lead to a representation which is quite noisy and problematic, although the Length-specific Attractors are formed using fewer demonstrations. Even more surprisingly, using a Length-specific Attractor of a different length compared to the prompted number still outperforms the generic Attractor, demonstrating once again that the "naive" averaging may land to a point in the latent space that does not correspond to a well-defined Attractor.

Table 3: Impact of Attractor in arithmetic operations (the parenthesis indicates the number of digits in the prompted number and in the calculated Attractor respectively).

|  | "-1" (2) | "-1" (3) | "-2" (2) | "-2" (3) | "+2" (2) | "+2" (3) |
|---|---|---|---|---|---|---|
| All-demonstrations Attractor | 96% | 58% | 66% | 19% | 93% | 51% |
| Length-specific Attractor (2) | **97%** | **61%** | **81%** | **27%** | **95%** | **56%** |
| Length-specific Attractor (3) | **98%** | **84%** | **68%** | **43%** | **93%** | **67%** |

