# OpenReview forum: "Concept Attractors in LLMs and their Applications"
_ICLR.cc/2026/Conference — ICLR 2026 Conference Desk Rejected Submission_

### Official Review · Reviewer_nRDX · 2025-10-26

**Soundness:** 2
**Presentation:** 3
**Contribution:** 3
**Rating:** 2
**Confidence:** 3

**Summary:**

This paper presents a novel conceptual framework for understanding the internal representations of Large Language Models (LLMs). The core hypothesis is that the observed convergence of semantically related, yet lexically diverse, prompts to similar representations at specific intermediate layers can be effectively modeled as an Iterated Function System (IFS). The authors posit that sequences of LLM layers act as contractive mappings, guiding representations toward concept-specific "Attractors."

**Strengths:**

The "Concept Attractor" analogy to IFS is a powerful and elegant conceptual tool. It provides a unified and intuitive explanation for several disparate, observed phenomena in LLM representation spaces, such as representational collapse and the existence of task vectors.

Moreover, the paper convincingly demonstrates that this single, simple idea can be successfully applied to a diverse and important set of problems (unlearning, safety, code generation, VLM grounding, data augmentation). This suggests the "attractor" is a general and fundamental property of these models.

**Weaknesses:**

I was one of the reviewers for this paper's previous submission (to NeurIPS).

In my previous review, I raised two main concerns:

1. The paper should explicitly describe the procedure used to identify attractors.

2. The paper does not provide compelling evidence that two attractors cannot converge to similar or nearby spaces within the same layer. This is because tasks like unlearning require not only editing the target mechanism but also ensuring that other mechanisms remain unchanged to combat fine-tuning issues like catastrophic forgetting.

However, I did not find the authors to have sufficiently addressed these two issues. In particular, regarding justification for downstream tasks, in addition to the aforementioned remaining issues, there is also a lack of comparison with recent SOTA work.

**Questions:**

see above

---

> ### Author Response · Authors · 2025-11-20
>
> Dear Reviewer,
>
> We are grateful for your engagement in both submissions, and appreciate the chance to address your concerns substantively.
>
> You will recall that at NeurIPS, the paper received a positive overall review. After a productive discussion, your last encouraging message was "... authors have addressed most of my concerns and I'd like to keep my positive rating." All reviewers maintained positive ratings of 4 (out of 6). Nothing major was noted in the AC summary.
>
> After the relatively unexpected NeurIPS decision, we had less than one week. Based on the positive post-rebuttal signal,  we incorporated most rebuttal content, expanded the related work, expanded to other LLM families (Figure 4), clarified the attractor identification procedure in multiple sections, and improved the presentation.
>
> Your current recommendation shows that we misunderstood the importance you placed on the formal specification of the algorithm or quantitative separation analysis. We appreciate that you like the idea/paper and want this included prominently. We will fix this. We apologize for the misunderstanding and hope to regain your vote of confidence. Below you can find our answers to all of your questions. Additionally, all changes have already been incorporated in the paper, which is available as a **revised version**.

---

> > ### Author Response · Authors · 2025-11-20
> >
> > ## Weakness 1: The paper should explicitly describe the procedure used to identify attractors
> >
> > We have now updated the manuscript so that we clarify the identification procedure in **Section 2.2** as well as in **Appendix A**.
> >
> > Regarding identification of the attractor layer, as discussed in Section 2.2, we simply select the layer exhibiting “maximum collapse” in all experiments. More specifically, our algorithm can be broken down into the following steps:
> >
> > 1. Estimation of each layer Attractors by averaging the hidden representations for the samples of each representative class/concept.
> > 2. Estimation of intra-distances, by finding the cosine similarity of each hidden representation from its corresponding attractor
> > 3. Estimation of inter-distances, by finding the cosine similarity of the Attractors
> > 4. Estimation of separation per layer, as inter - intra.
> > 5. Finding the optimal layer, i.e., the layer that maximizes separation.
> >
> > ## Weakness 2: Evidence that two attractors cannot converge to similar or nearby spaces
> >
> > We understand your concern. Simply hoping for separation is not good enough; we should understand if there is some mechanism behind it. We believe the answer lies in working backward from the empirical performance of these models. All of us acknowledge that we get good performance across diverse tasks. Let us consider what this implies for the internal state.
> >
> > A sensible premise is that the model successfully generates sensible textual distributions for distinct concepts (e.g., JK Rowling versus Tolstoy). Now, the generation process from the attractor layer, say $L$, to the final vocabulary output is governed by the starting point at L and the work done by the subsequent layers ($L+1$ through to the output). Let us denote this "decoding" function as $D()$. But the layers are continuous functions. If the attractors for Rowling ($h_R$) and Tolstoy ($h_T$) had converged to the same or very nearby spaces around Layer $L$, then due to the continuity of $D$ the outputs would be difficult to tell apart. i.e., $D(h_R)$ close to $D(h_T)$. So, if the model is generating distinct styles, the inputs to the decoding function (attractors/fixed points) must be geometrically separate. If they were not, we would be getting indistinguishable outputs. If we view pre-training (up to the attractor layer) as figuring out how to map text to the correct attractor, and if the concepts overlapped, the confusion between Rowling and Tolstoy would be high. There is an incentive to have a margin sufficient for the decoder to map them correctly.
> >
> > Regarding the reviewer’s specific question about unlearning, the Utility metric is indeed measuring exactly that (Figure 6 and Appendix C). The Utility score checks/quantifies whether modifying one concept's attractor (e.g., removing T.S. Eliot) impacts the model's performance on related but distinct concepts (e.g., other authors or general literary knowledge). Our results show minimal utility degradation, indicating that: (a) concept attractors are sufficiently separated to allow selective manipulation, and (b) our intervention can target specific concepts without significant collateral damage or impact on nearby concepts. We appreciate the suggestion, and we explicitly mention that now in **Section 3** of the main paper.
> >
> > Additionally, our code translation experiments show that modifying one programming language's representation doesn't interfere with the model's understanding of other languages. These results provide empirical evidence that concept attractors maintain sufficient separation for practical applications beyond unlearning, too.
> >
> > Finally, following Weakness 1, we show the separation between different Attractors in **Appendix A**. From these results, we can observe that the Attractors of different Authors are well separated in the layer of choice, minimizing the risk of “accidental removal” or “catastrophic forgetting”.

---

> > > ### Author Response · Authors · 2025-11-20
> > >
> > > ## Comparison with additional application-specific SOTA methods
> > >
> > > At your suggestion, we have now expanded our comparisons to include additional recent SOTA works across the main applications. While our contribution is to give a unified strategy to address all these diverse tasks with a single scheme, rather than optimize for a specific application, our approach is quite competitive with or exceeds specialized methods while requiring no training and no retention data.
> > >
> > > A) Unlearning (TOFY)
> > >
> > > - ALKN (ICML 2025, https://openreview.net/pdf?id=tcK4PV3VN4): Training-based. It achieves a Utility value in the range (50, 55).
> > > - LUNAR (arxiv - Oct 2025, https://arxiv.org/pdf/2502.07218): Achieves a maximum Utility score of 60.8.
> > > - Ours: Our approach maintains a value above 61 in all of the TOFU-related experiments (≥ 61.20). It is also training free.
> > >
> > > B) Code translation (Python → JS)
> > >
> > > - RFM (arxiv - May 2025, https://arxiv.org/pdf/2502.03708): Achieves a result of 4.34 in the task of Python to JavaScript translation.
> > > - Ours: Simpler and train-free. Achieves a value of 4.56.
> > >
> > > C) Hallucination reduction (VLMs)
> > >
> > > C.1) Llava 1.5
> > >
> > > - VisVM (ICCV 2025, "Scaling Inference-Time Search with Vision Value Model for Improved Visual"): Achieves CHAIRs result of 31.7 and a result of 11.9 in CHAIRi, while it also results in a more than 10x increase in the inference time.
> > > - IBD (CVPR 2025, "Alleviating Hallucinations_in Large Vision-Language Models via Image-Biased Decoding"): Requires using an auxilary LVLM, i.e., a 2x increase in memory demands and an at least 2x increase in inference time. Achieves CHAIRs and CHAIRi of 25.6 and 10.2, respectively.
> > > - Ours: CHAIRs 36.2, CHAIRi 9.8, while maintaining the same throughput as the original model.
> > >
> > > C.2) InstructBLIP
> > >
> > > - IBD (CVPR 2025, "Alleviating Hallucinations_in Large Vision-Language Models via Image-Biased Decoding"): Requires using an auxilary LVLM, i.e., a 2x increase in memory demands and an at least 2x increase in inference time. Achieves CHAIRs 18.2 and CHAIRi 7.1 in InstructBLIP.
> > > - Ours: CHAIRs 19.7, CHAIRi 6.9, with no compute overhead.
> > >
> > > Our approach achieves comparable performance to recent specialized methods across all these applications while offering: (1) zero training required, (2) no retention data, (3) no compute overhead at inference, and (4) a single principle works across diverse tasks. The fact that one idea is practical and remains competitive with strong, optimized, task-specific methods is the main value of our work.

---

### Official Review · Reviewer_xGFZ · 2025-10-28

**Soundness:** 3
**Presentation:** 3
**Contribution:** 3
**Rating:** 6
**Confidence:** 3

**Summary:**

The paper proposes a novel perspective on the phenomenon of representation convergence in LLM, hypothesizing that the internal dynamics of LLMs can be understood through the framework of Iterated Function Systems. Specifically, it suggests that layers in an LLM act as contractive mappings, driving semantically related prompts (even with diverse lexical forms) to cluster around concept-specific Attractors in LLM's latent space. These attractors are invariant sets that characterize specific concepts (e.g., "Harry Potter" or "Python programming").

The core contribution is leveraging this "Attractor" concept to develop simple, training-free, and data-efficient interventions that operate directly on these Attractor vectors to solve a diverse range of practical tasks including hallucination reduction, guardrailing, and synthetic
data generation.

**Strengths:**

1. The hypothesis that LLM layer transformations act as contractive mappings converging to concept-specific Attractors that can be modeled by Iterated Function Systems is novel and an interesting perspective.
2. Beyond the observation, I like the practical applications explored in this paper. Though the intervention itself is simple in its form, addition / subtraction, but its effectiveness and generalizable observations in many downstream tasks is a major strength.
3. The paper is well-structured and clearly written.
4. Wide experiments with model variety, LLM and VLM, various datasets are used for evaluation.

**Weaknesses:**

1. While the IFS hypothesis is the conceptual backbone, the actual modeling is simplified to a single affine contractive map $\phi_{eff}$ whose unique Attractor is a single fixed point $V^*$. This 1-map model is acknowledged as a "first-order approximation". The paper doesn't fully demonstrate a more complex, multiple-map IFS model needed for concepts with "complex geometries" or fractal structures (like those observed in Figure 4 left, e.g., numerical relationship between 1/2/3 digits). This leaves a gap for the simplified model used in practice (1-map affine transformation). To strengthen the paper, a minimal example of a multi-map IFS fit or a deeper analysis of the fractal dimension/geometry would be strong.

2. The concept is defined informally (e.g., "Lord of the Rings universe," "Python programming language," "toxic language"). It is also noted that the same concept can have multiple Attractors (e.g., English text, Python style, multi-digit arithmetic), which raises questions about the robustness of the "Attractor" concept and the manual effort required to determine the appropriate layer and the boundary of a "concept" (i.e., when should Python's OOP and procedural styles be treated as one or two Attractors?). This is actually somehow related to my Weakness 1, which may also offer a more flexible and compex framework for the downstream application.

3. It is a bit blur why the simple vector arithmetic would work for various downstream task. I understand the author provides a set of tasks and experiments to demonstrate the validity of the method. Substracting the Attractor vector for detoxification is somehow intuitive, but why simply adding a language Attractor (e.g., Python to Javascript) not interfere the code context but translate to another programming language? In some tasks, it is quite tricky to define the Attractor and find the operation need to be done.

**Questions:**

1. Related to Weakness 3, the paper states that Attractors form at different layers for different concepts (e.g., fiction at L24, programming at L19, natural languages at L27). What is the hypothesis or metric used to determine the optimal Attractor layer for an arbitrary new concept C? Is it always the layer where the inter-prompt distance is minimized (as suggested in Section 2.1)? If the selection process requires experimentation, this slightly compromises the "training-free" and efficiency claims for new concepts. Can the authors formalize the process for finding the most "contracted" layer $l_C$?

Other questions are discussed as the details in Weakness.

---

> ### Author Response · Authors · 2025-11-20
>
> We thank the reviewer for their time and effort. Below, we analyze each of their questions. We hope this will help clarify any remaining concerns about our work and strengthen the support for it. We would like to mention that all changes are available as a **revised version**.
>
> ## Weakness 1: multi-map IFS examples
>
> We appreciate the constructive suggestion. While the single-map model is sufficient for the tasks in Sections 3–5, your comment asked us to explore richer IFS structures. Motivated by your observation of fractal-like patterns (Fig. 4), we investigated whether these reflect multiple attractors requiring a multi-map IFS model.
>
> Following your suggestion, we analyzed the formation of multiple attractors in arithmetic tasks. Consider steering the model to perform “subtract 2’’ directly in latent space, using only a number prompt “i→’’ (with i substituted). In principle, one can add the “–2’’ attractor at the appropriate layer. However, as visible in Fig. 4, multiple “–2’’ attractors form depending on the digit count of the demonstrations—suggesting several sub-attractors rather than one fixed point.
>
> We compared two variants (using “–2’’ as an example):
> 1. Add the “–2’’ attractor computed from all demonstrations, regardless of digit count (1-map model).
> 2. Add the “–2’’ attractor computed only from demonstrations with the same digit count as the input number (multi-map model).
>
> As you anticipated, the results favor the multi-map structure. For single-digit numbers the data are too limited, but for 2- and 3-digit numbers we see up to a 25% absolute improvement when using the digit-specific attractor over the more general (and noisier) one.
>
> **Appendix F** provides full details. This shows that averaging over demonstrations can be problematic when multiple smaller attractors exist (e.g., digit-count clusters), exactly as you suggested. In short, concepts with hierarchical structure naturally require multi-map IFS models: each digit-count cluster behaves like its own contractive map with its own fixed point, creating the fractal-like structure in Fig. 4. Our 1-map approximation works well for concepts with a single well-defined attractor but can be limited when internal substructure is present. We are grateful for the suggestion and plan to pursue this direction more broadly in future work.
>
> ## Weakness2/Question 1: What is the process of obtaining the Attractor’s layer
>
> As the reviewer correctly points out, the layer we select is the one that maximizes inter-cluster distances while maximizing intra-cluster compactness. We appreciate the suggestion, and we have detailed the procedure in **Appendix A**. We want to emphasize that no other experimentation and/or training is required, hence the train-free nature of our approach.
>
> More specifically, our algorithm can be broken down into the following steps:
>
> 1. Estimation of each layer Attractors by averaging the hidden representations for the samples of each representative class/concept.
> 2. Estimation of intra-distances, by finding the cosine similarity of each hidden representation from its corresponding attractor
> 3. Estimation of inter-distances, by finding the cosine similarity of the Attractors
> 4. Estimation of separation per layer, as inter - intra.
> 5. Finding the optimal layer, i.e., the layer that maximizes separation.
>
> ## Weakness 3: Why vector arithmetic works for downstream tasks
>
> Vector arithmetic appears to work because attractors represent stable configurations the model converged to during training, and operations between them capture systematic relationships. The empirical results suggest that attractors, as fixed points of the model’s dynamics, occupy geometrically meaningful positions.
>
> In code translation, Python and JavaScript form distinct attractors (Fig. 4). Their difference vector reflects how the model represents the two languages. Adding this vector to a Python code representation steers the hidden state toward a configuration the model interprets as valid JavaScript, after which later layers resolve and decode it into coherent output. The intervention only needs to push the state into a reasonable JavaScript basin: once initial JavaScript tokens appear at timestep t, they enter the context for t+1, and the model’s forward dynamics sustain further JavaScript generation. The edit need not be perfect, just sufficient to move the state into a recognizably JavaScript-like region. If attractors were geometrically unstructured, such vector arithmetic would fail, so our experiments indicate that attractor locations are meaningfully organized fixed points of the learned dynamics.
>
> We agree, however, that the deeper reason this geometric organization emerges during training remains open. The IFS analogy suggests semantically related inputs converge to fixed points, but this alone does not explain why arithmetic between concept attractors should preserve coherence. This remains a limitation of our current understanding.

---

### Official Review · Reviewer_C7VY · 2025-10-29

**Soundness:** 2
**Presentation:** 2
**Contribution:** 2
**Rating:** 4
**Confidence:** 4

**Summary:**

This paper reveals that Large Language Models (LLMs) map semantically related prompts to similar internal representations at specific layers, regardless of surface form. The authors model this convergence using Iterated Function Systems (IFS), where LLM layers act as contractive mappings toward Concept Attractors that are specific to high-level concepts. Leveraging this insight, the paper proposes a suite of simple, training-free methods that manipulate these attractors directly to address diverse practical applications, including language translation, hallucination reduction, safety guardrailing, and synthetic data generation, achieving performance comparable to or exceeding complex, specialized baselines with minimal computational cost.

**Strengths:**

1. Novel and Insightful Theoretical Framework: The paper introduces the Iterated Function Systems (IFS) and Concept Attractors as a compelling dynamic systems explanation for the phenomenon of representational convergence in LLMs, providing a new, high-level lens for mechanistic interpretability.
2. Highly Efficient and Versatile Training-Free Intervention: The work develops and validates a set of straightforward, zero-cost (training-free) methods based on direct attractor manipulation. This offers a highly practical and general solution for a broad spectrum of tasks (e.g., translation, safety) that avoids the computational expense and complexity of fine-tuning or specialized model architectures.
3. Strong Performance with High Practical Value: Despite their simplicity, the proposed attractor-based methods achieve performance competitive with or better than complex, dedicated fine-tuning baselines. This demonstrates significant utility, particularly for resource-constrained environments or scenarios requiring rapid, on-the-fly model steering.

**Weaknesses:**

1. While the IFS analogy is powerful, the formal connection remains somewhat hypothetical and is justified more by empirical observation than by a rigorous theoretical proof. The paper does not formally demonstrate that transformer blocks satisfy the conditions for being contractive mappings in a way that guarantees convergence to a unique attractor as defined in IFS theory. The 1-map affine approximation (Eq. 3) seems like a significant simplification.

2. The method for estimating an attractor (averaging activations) is simple, but its robustness is not deeply explored. More critically, the paper notes that attractors form at different layers for different concepts (e.g., L19 for code, L24 for fiction). The paper lacks a clear, systematic methodology for identifying this crucial "attractor layer" for a novel, arbitrary concept. This ambiguity is a significant barrier to applying the method in new domains.

**Questions:**

question:
1. Could you please elaborate on the methodology for identifying the correct "attractor layer" for a new concept? Is there a systematic or automated way to find the layer of maximal "collapse" (e.g., by tracking the inter-prompt distance from Eq. 1 across layers) without an exhaustive, manual search? This seems to be the most critical hyperparameter for your method.

2. The paper simplifies the attractor to its mean vector. However, the t-SNE plots and the discussion of fractal structures suggest these attractors are regions or manifolds with complex geometries. Have you explored interventions that use a more complex representation (e.g., modeling the attractor as a subspace via PCA, or as a Gaussian distribution) rather than a single point-mass average?

---

> ### Author Response · Authors · 2025-11-20
>
> We thank the reviewer for their time and effort. Below, we analyze each of their questions. We hope this will help clarify any remaining concerns about our work and strengthen the support for it. We would like to highlight that all changes have already been incorporated in the paper, which is available as a **revised version**.
>
> ## Weakness 1: The IFS analogy is hypothetical; a theoretical proof is missing
>
> We thank the reviewer for finding our work novel and insightful. We are glad the IFS analogy resonated, and we clarify a few points. We do not claim that transformer layers are formally contractive mappings satisfying IFS convergence theory, nor that a single-map affine approximation captures the full dynamics. Our point is that an IFS-like view (layers progressively collapsing semantically related representations) offers a useful framework that explains observed behavior and yields effective interventions across tasks that have previously been studied in isolation.
>
> A formal proof that transformers implement an IFS would require extreme simplifications that limit practical value. Establishing contractivity would require bounding operator norms across attention, layer norm, nonlinearities, and residual connections under the full complexity of learned weights and input distributions. Input dependence through softmax and layer-norm scaling further complicates this. Even if each component were bounded, their composition through residuals forms a coupled system that must be contractive for all inputs and all learned parameters; something unavailable even for much simpler architectures.
>
> Much prior interpretability work relies on empirically validated analogies rather than formal proofs: induction heads as algorithmic pattern matching, task vectors as linear decompositions, belief-state geometry as probabilistic inference. These remain useful frameworks despite the lack of formal guarantees. Our work is similar: a framework/analogy rather than a theorem. We will update the text to reflect this more clearly.
>
> ## Weakness 2/Question 1: Automated discovery of the Attractor layer
> Our method for identifying attractor layers is a simple algorithm (detailed now in **Appendix A**):
> 1. For each layer, estimate attractors by averaging hidden representations within each concept.
> 2. Compute intra-cluster distances (representations to their attractor, using cosine similarity).
> 3. Compute inter-cluster distances (distances between attractors).
> 4. Compute the separation score: inter minus intra.
> 5. Choose the layer with maximal separation.
>
> This requires no hyperparameter tuning, no training, and no expertise beyond defining “semantically related’’ prompts. Standard clustering metrics would work as well.
>
> In practice, collapse is dramatic and visually clear: distance plots (Figures 4–5) show a sharp drop at the attractor layer. For a new concept, it is a simple as collecting a small set of semantically related examples (≈20–100), running forward passes to get hidden states, computing inter-prompt distances, and locating the layer where collapse is maximal.
>
> ## Weakness 2/Question 2: Estimation of attractor beyond averaging
>
> This is a good question. Under the IFS analogy, point-mass averaging is the correct choice. A contractive affine map $\phi(x)=Mx+b$ with $|M|<1$ has a unique fixed point $x^*$ that all trajectories converge to, so an attractor is a single point in latent space, not a distribution or subspace. PCA subspaces or Gaussians instead model the basin of attraction or noise from incomplete convergence, not the attractor itself. The spread in t-SNE reflects imperfect convergence and sub-concept structure, not the attractor; arithmetic tasks show trajectories approaching multiple attractors (e.g., one per digit count), not a complex one. By the attractor layer (e.g., in the fictional-worlds example), representations collapse to tight clusters consistent with point attractors. The remaining variance is likely noise: PCA captures this stochasticity, whereas averaging cancels it to estimate the fixed point.
>
> Empirically, simple averaging outperforms alternatives: our point-based attractor operations match or outperform ICV and RFM on toxicity and code translation, indicating that the fixed point is the right object for these tasks. Section 5 shows that multiple instructions for the same sample yield a common representation, demonstrating attractor formation even under prompt variation; richer structure would cause these single-sample interventions to fail. Some scenarios may still benefit from extensions: when concepts split into sub-concepts, identifying multiple attractors can help (Appendix F). These are refinements for edge cases, not limitations of the main approach.

---

### Official Review · Reviewer_wEEx · 2025-11-01

**Soundness:** 3
**Presentation:** 3
**Contribution:** 2
**Rating:** 6
**Confidence:** 2

**Summary:**

This paper presents the idea that Large Language Models (LLMs) internally organize information using "Concept Attractors". The core phenomenon observed is that semantically related prompts, even with very different wording (e.g., "Who is Gandalf the Grey?" vs. "What is the significance of Mount Doom?"), are processed by the model's layers until their internal representations "collapse" into nearly identical locations in the hidden state.

The authors model this behavior through the lens of Iterated Function Systems (IFS), suggesting that the sequence of transformer layers acts as a set of contractive mappings. These mappings progressively guide any input related to a specific concept (like Lord of the Rings or the Python programming language) toward a unique, stable Attractor. The paper empirically shows that these Attractors form at different layers for different concepts and can even exhibit complex, fractal-like structures.

The main contribution is demonstrating that these Attractors can be directly manipulated using simple, training-free interventions. By estimating a concept's Attractor (often by just averaging the hidden states of a few examples ), the authors can steer the model's behavior. For instance, they reduce toxicity by subtracting the "toxic Attractor" from activations and perform code translation by adding the target language's Attractor to steer the generation . This "Attractor-based intervention" is also applied to machine unlearning (by guardrailing responses that get too close to a "forget" Attractor) , to reduce hallucinations in vision-language models by reinforcing the "visual Attractor" during generation , and to create more factual, diverse synthetic data by perturbing Attractors. These simple, computationally efficient methods are shown to match or exceed the performance of specialized, resource-heavy baselines.

**Strengths:**

- The paper introduces an interesting theoretical framework to understand the phenomenon of "clustering" across semantically similar prompts.

- The interventions the paper proposes are very lightweight from a computational point of view.

- The experimental results (on the small-scale models) are compelling.

**Weaknesses:**

- This method, while computationally easy to implement, requires white-box access to the model.

- The experiments are done on models of relatively small scale.

- The model feels a bit simplistic, I'm not sure if the hypothesis of concept attractors generalizes to more complicated real-world tasks.

**Questions:**

- What conclusions from your experiments do you think would generalize to more complicated tasks and models?

- Do you have any suggestions of how one can take advantage of these observations you have in your paper when given black-box access to an LLM? Would some intermediate type of access (e.g., through log-probabilities) be useful?

---

> ### Author Response · Authors · 2025-11-20
>
> We thank the reviewer for their time and effort. Below, we analyze each of their questions. We hope this will help clarify any remaining concerns about our work and strengthen the support for it. We would like to mention that all changes are available as a **revised version**.
>
> ## W1/Q2: White-box assumption
>
> Yes, our approach needs white-box access, but given current deployment realities as well as regulatory requirements, this is not very restrictive.
>
> a) White-box access is common when organizations deploy open-weight models (Qwen, DeepSeek, Mistral) on their own hardware or cloud (AWS Sagemaker, Google Vertex). The gap between closed and open access models is becoming narrower, so the overall scope of where white box access is reasonable will continue to grow. So there is broad applicability.
>
> Yes, major API providers (OpenAI, Gemini, Anthropic) offer only black-box access to users like us. But these providers must implement equivalent interventions internally for regulatory or other reasons, which cannot be satisfied with prompt-level filtering alone. Our unlearning work shows a simple mechanism a provider will need or can use. So, the methods are applicable to provider-side implementations independent of what the external APIs expose.
>
> b) Regarding log-probability alternatives, one difficulty we find with it, is that they only expose the final vocabulary projection, discarding the layer-specific attractor dynamics that our method needs. In our experiments, layer-specificity turns out to be important because different concepts stabilize at different depths. So, a strong intervention requires targeting the correct layer. Exploring whether log-probability patterns plus some clever prompt sampling could provide indirect signals about attractor formation will be an interesting endeavor if one can get it to work well. We appreciate suggestions or ideas!
>
> ## W2/Q1: LLMs are small-scale
>
> Yes, our experiments focused on 7-9B parameter models rather than frontier models. This choice was based on 1) computing budget constraints, 2) the existence of only 7B models trained on TOFU, and 3) maintaining a fair comparison with existing (7B-based) baselines.
>
> Empirically, we showed consistency across two distinct model families (Llama, Qwen, see Fig. 4) with different architectures. We see attractors manifesting at specific layers for different concepts in a reproducible way across architectures. These cross-family experiments give us much confidence regarding the generality of our findings that this is indeed a more general property of transformer-based LLMs.  None of our experiments gave us any signal that the behavior was dependent on the model or model size. Related work, such as task vector formation,n has also been observed across different model scales and different model families without degradation.
>
> We welcome the constructive suggestion. We have expanded our appendix (**Appendix B**) to show the formation of Attractors on 3B and 14B models too, covering, based on your comment, a wider selection of different LLM sizes. Our observations also hold in these models.
>
>
> ## W3/Q1: Main takeaways for generalization to more complex tasks
>
> Thanks for the question! One main generalizable insight is that LLMs appear to perform semantic contraction, collapsing surface-form variations into concept-specific attractors at intermediate layers. This appears to be a property of how transformers process information. The practical implication is that the model has already done the "hard work" of organizing its knowledge geometrically; if we can just identify and manipulate the right locations, we can use the idea for numerous benefits. Simple geometric interventions (addition, subtraction, maintenance) on collapsed representations outperform complex training-based methods.
>
> For complex reasoning tasks, we suspect that attractor formation almost certainly occurs, but perhaps in a more sophisticated form. Consider multi-hop reasoning. Rather than converging to a single static attractor, the model can perform attractor chaining, where it performs sequential traversal through intermediate concept attractors for reasoning steps. For example, answering "Who wrote the book that inspired the movie Lincoln, and who starred in it?" might involve: Query -> [Author attractor] -> [Book attractor] -> [Movie attractor] -> [Actor attractor] -> Answer. Each reasoning hop could correspond to transitioning between attractors in latent space. Interventions for reasoning tasks would not steer toward/away from a single attractor, but rather facilitate transitions between attractors. The core principle that semantic structure exists geometrically in latent space and permits manipulation would still hold, but the intervention strategy of attractor chains would be much more involved. In this work, we have verified the idea for static concepts, and extending it to dynamic reasoning will need additional development for "temporal" attractor dynamics.

---

### Author Response · Authors · 2025-11-27

Dear Reviewers,

We would be happy to provide any additional clarification or address any further concerns regarding our submission. We hope that our initial responses, together with the revised version of the paper, have helped resolve the points you raised. Please let us know if any further information is needed.

---

### Note · Program_Chairs · 2026-01-19
**Submission Desk Rejected by Program Chairs**

The updated paper draft contains author names, violating double blind. The submission must be desk rejected.